# COVID-19-Related Awareness and Behaviors of Non-Saudi Travelers and Their Satisfaction with Preventive Measures at Saudi Airports

**DOI:** 10.3390/tropicalmed7120435

**Published:** 2022-12-13

**Authors:** Aseel Ali AlSaeed, Unaib Rabbani, Abdulrhman Aldukhayel, Sultan Alzuhairy

**Affiliations:** 1Family Medicine Academy, Qassim Health Cluster, Buraidah 52385, Saudi Arabia; 2Department of Family and Community Medicine, College of Medicine, Qassim University, Buraidah 51452, Saudi Arabia; 3Department of Ophthalmology, College of Medicine, Qassim University, Buraidah 51452, Saudi Arabia

**Keywords:** awareness, behaviors, COVID-19, travelers, preventive measures, Saudi Arabia

## Abstract

Airports could serve as hotspots for the spread of the COVID-19 infection. We aimed to assess the awareness, attitude, and behaviors of non-Saudi travelers about COVID-19 and their satisfaction with preventive measures at Saudi airports. A cross-sectional study was conducted among non-Saudi travelers arriving in Saudi Arabia at two international airports. Data were collected using a questionnaire in Arabic, Hindi, and Urdu languages. Awareness, attitude, behavior scores, and satisfaction levels were calculated. Linear regression analyses were done to assess the predictors of awareness, attitude, and behaviors. A total of 633 participants were included in the study. The mean awareness, attitude, and behavior scores were 16.3/22, 18.3/24, and 3.2/5, respectively. Nationality, education, and income were significant predictors of awareness. Nationality was the only significant predictor of attitude, while gender and nationality were predictors of behaviors. Overall satisfaction with preventive measures was: transportation (94%), lounges and corridors (96%), counters (98%), and airport staff (97%). There was overall good awareness and attitude among travelers. Behavior scores were relatively low, which needs to change as air travel poses a threat to the spread of infectious diseases. Airport authorities need to manage passengers properly to ensure adequate distancing to prevent the potential transmission of infections.

## 1. Introduction

Coronavirus disease 2019 (COVID-19) was declared a pandemic on 11 March 2020 [1]. Since then, peoples’ lives have changed in many ways such as threats to health, social disruption, and economic losses [2,3]. The number of confirmed COVID-19 cases mounted to more than 635 million and total deaths were more than 6.5 million as of 15 November 2022 [4]. In order to control the pandemic, countries took intensive measures such as social distancing, lockdowns, and restrictions of social and economic activities [5]. The direct morbidity and mortality due to infection, along with social and economic restrictions, caused widespread psychosocial effects among populations globally [3,6].

The tourism and hospitality industries were the earlier sectors affected by this pandemic [7]. The volume of customers in the aviation industry decreased by 70–80% during April 2020 compared with April 2019 [8]. Because of its worldwide spread, the Saudi government took very strict measures, including suspending schools, quarantining all over the country, and minimizing social activities. In addition, from the start of the first quarter of 2020 until 17 May 2021, the Saudi authorities announced travel restrictions on residents and citizens from traveling aboard [9,10].

With the rollout of vaccinations against COVID-19 and control of the infection’s spread, countries gradually eased the travel restrictions. This would make the ports of entry hotspots for the further spread of the disease. This holds true not only for COVID-19 but also for other diseases; the latest one being monkeypox. Therefore, during the next stage of containing the virus, travelers’ awareness, attitudes, and behaviors toward the disease would be important determinants of the spread of infection. The impact of measures taken by the governments largely depends on the awareness and behavior of the population [11]. For this reason, it is essential to explore the travelers’ awareness and behaviors toward this disease. However, most studies have targeted the general population and health professionals [12,13,14]. These studies have also shown various socio-demographic factors such as age, gender, education, and residence associated with awareness and behaviors related to COVID-19 [12,13,14,15]. Studies from Saudi Arabia have also reported a good level of knowledge and better practices related to COVID-19 in the general population [9,16,17,18].

The COVID-19 outbreak is considered the first pandemic that globally affected air transport drastically [19]. Travel intentions during the COVID-19 pandemic were found to be affected by several factors related to airlines and airports. A study investigating the effect of COVID-19 global travel restrictions on Saudi tourists revealed that their behavior, habits, and travel preference were not the same as before the COVID-19 pandemic [10]. Transport safety became the most important priority among travelers and the essential part was preventive measures [10]. The same was noticed in a study conducted in Bulgaria, where older people showed a lower preference for air transport [7]. A study done among travelers in Sharam ElSheikh airport showed personal protective measures and sanitization procedures had a positive influence on the intention to travel [20]. Another study revealed that the majority (75.6%) of travelers showed somewhat extreme concerns about being infected during flying [21]. As part of infection control measures during air travel, most of the passengers in a study agreed that it is very important to provide complementary hand sanitizer, face masks, and sanitary wipes to the passengers, as well as provide them with information regarding the ways to prevent the spread of infection [21]. The most significant factors that may impact the decision for choosing a destination among populations of the Middle East were infection control and preventive measures [7,22].

The effect of safety procedures is an essential variable on passenger satisfaction and airport performance [20]. “Cleaning”, as a factor for achieving passenger satisfaction, increased significantly after the COVID-19 pandemic [23,24]. Before the pandemic, most of the satisfaction was attributed to restroom facilities and drinking water [8]. Information about infection control during travel is an important factor. A study revealed that most participants, when asked about their satisfaction with the information, were unsure (73.1%) and dissatisfied (18%) [21]. As the COVID-19 pandemic is coming under control with waves in different parts of the world, the aviation industry is back on resuming its operations and most travel restrictions have been eased. It is important to know the awareness and behaviors of travelers regarding the disease and their satisfaction with infection control measures at airports. This information is relevant not only for the COVID-19 pandemic but also for other global threats such as monkeypox in the future.

We, therefore, aimed to assess the non-Saudi traveler’s awareness of COVID-19 and their attitudes and behaviors toward it. In addition, we also measured travelers’ satisfaction with the infection control measures at airports.

## 2. Materials and Methods

### 2.1. Study Design and Setting

A cross-sectional study was conducted at King Khalid International Airport, Riyadh, and Prince Naif Bin Abdulaziz Airport, Qassim, Kingdom of Saudi Arabia (KSA) among non-Saudi travelers returning to Saudi Arabia from 10 October 2021 to 17 March 2022. King Khalid International Airport, Riyadh is the second largest airport in KSA, with an estimated 3.79 million international travelers in 2019, while Prince Naif Bin Abdulaziz Airport, Qassim is the fifth largest airport, with about 174,000 international travelers in 2019.

### 2.2. Sample Size

The sample size was calculated using the WHO sample size determination in health studies. Since no such study has been conducted among travelers previously, we assumed at least 50% of the travelers to be satisfied with infection control measures at airports. At a 95% confidence level with a 5% margin of error and a design effect of 1.5, the sample size calculated was 577 participants. For estimating the means scores of awareness, attitude, and behaviors, we used estimates from a previous study conducted in Saudi Arabia that used similar tools to assess these outcomes as a reference [17]. At a 95% confidence level and a margin of error of 2% around the population mean, the largest sample size calculated was 455 participants based on a mean and standard deviation of the practice of 4.34 (±0.87). To achieve these two objectives, we needed at least 577 participants in our study.

### 2.3. Sampling Procedure

Convenience sampling was used to recruit participants from waiting areas outside the arrival terminals and parking lots of selected airports. Once travelers finished all the official checks and collected their luggage, they were invited to participate in the study. Two trained data collectors explained the purpose and procedure of study to the potential participant and requested their voluntary participation. Once the voluntary consent was given, the data collectors assessed them for eligibility criteria. Non-Saudi travelers entering Saudi Arabia either through direct flights from their countries or transiting for less than 24 h in other countries were eligible to participate in the study. Participants were excluded if they had confirmed COVID-19 infection previously. We included only non-Saudi travelers based on the assumption that Saudi nationals would constitute the very small proportion of arriving travelers as traveling outside the KSA was banned for citizens until late May 2021. Those who were stuck outside KSA were brought back early during the pandemic through special flights. This might have resulted in under sampling of the nationals relative to their population size in the country and would not give any reliable estimates pertaining to the nationals.

### 2.4. Study Tools

Data were collected using a structured questionnaire in three languages; Arabic, Hindi, and Urdu. The questionnaire had five sections. The first section was about socio-demographics that included age, gender, marital status, level of education, nationality, employment status, occupation, and reason for travel. The second, third, and fourth sections included variables on awareness (22 items related to the source, transmission, clinical features, risk factors, and prevention), attitude (6 items on social distancing, handwashing, and controllability of pandemic), and behaviors (5 items related to social distancing, physical contact, and handwashing) of travelers about COVID-19. Variables in these sections were derived from the literature [16,17,25]. The last section assessed the satisfaction of travelers with preventive measures for COVID-19 at airports. Domains in this section included satisfaction with airport transportation (5 items), lounges and corridors (8 items), counters (5 items), and airport staff (6 items). The questionnaire was first translated into Arabic, Urdu, and Hindi languages by respective bilingual experts. In the next step, the translated versions were back-translated into English by another set of bilingual experts to assess the accuracy of the translation.

### 2.5. Data Collection Procedure

Data were collected by trained collectors using the questionnaire-based interviews. After the informed consent was obtained, the data collectors administered the questionnaire in the respective language of the participants. Data collectors were instructed to follow the exact phrasing and sequence of questions as written in the questionnaire in order to ensure standardized data. Participants were requested to provide accurate answers to each of the questions. Furthermore, respondents were assured of confidentiality and the non-judgmental approach of the research team towards any of their responses.

### 2.6. Data Management and Analyses

Data were entered into Microsoft Excel and then exported to SPSS version 21.0 (IBM SPSS Statistics for Windows, Version 21.0. Armonk, NY, USA: IBM Corp.) for analysis. Descriptive analysis was done to calculate frequencies and proportions of categorical variables and means along with standard deviations for continuous variables. For the questions related to awareness a score of “1” was assigned if the answer was correct and “0” if the answer was incorrect. All 22 items were then summed up to calculate the total awareness score. The score for each of the items in the attitude section ranged from “0” strongly disagree” to “4” strongly agree. A sum of all items in the attitude section was conducted to calculate the total score. In the behavior domain, “0” was assigned for poor behavior and “1” was assigned for good behavior. Subsequently, the total score of behavior was calculated after summing all the items. For bivariate comparisons of awareness, attitudes, and behaviors, we used an independent sample *t*-test, ANONA, and Pearson correlation. Univariable and multivariable linear regression analyses were carried out to calculate crude and adjusted beta coefficients along with associated 95% confidence intervals. Variables in the final regression model were included based on *p*-value cut-off of 0.25 in the univariable analyses and biological plausibility. Multicollinearity among the predictor variables was assessed using Kendall’s τ_b_ Cramer’s V and phi for categorical variables, and correlation (Pearson and Spearman) for continuous variables [26]. The cut-off used to remove variables based on multicollinearity from the multivariable model was 0.5 [27]. A *p*-value less than 0.05 was considered significant.

### 2.7. Ethical Considerations

The proposal for this study was reviewed and approved by the Qassim Regional Bioethics Committee (approval number #1442-531746, dated: 2 November 2020). No personal identifiers such as name, ID or passport number, or phone number were collected. The safety of the participants and data collectors was also ensured by training data collectors about safety measures and maintaining a safe distance of about two meters from the participants. Data collectors were also provided with surgical face masks, disposable gloves, and hand sanitizers.

## 3. Results

A total of 633 individuals participated in this study. Three-quarters (76%) of them were males The mean age of participants was 36 (±8.7) and ranged from 19 years to 63 years. The majority (86%) were married. The majority of the participants were either Indian (55.6%) or Pakistani (34.0%), followed by Egyptian (3.8%), Bangladeshi (3.6%), and Sudanese (3.0%). One-third (33%) of participants had a university degree. Most of the participants (77%) were employed in the private sector. A little less than half (44%) had an income of less than SAR 3000 per month. The majority of participants (76%) reported “Return to Work” as a reason for travel.

The mean awareness score of participants was 16.3 (±2.86) on a scale of 22. The mean score (±SD) of the attitude domain was 18.3 (±2.90) out of 24. The mean score for the behaviors was 3.2 (±1.04) out of 5. Table 1 presents the differences in awareness, attitude, and behavior scores of travelers to Saudi Arabia. We found that there was no significant correlation between age and awareness and attitude scores. However, there was a weak negative correlation between age and behavior scores. Females had significantly higher scores than males in awareness and attitude domains. Awareness and behavior scores were significantly higher for Indian and Pakistani nationals compared with others, while attitude scores were lower. There was a gradual increase in the scores in all three domains with increasing educational level. The high-income group had significantly higher awareness scores, while there was no significant difference in attitude and behavior scores.

Table 2 presents the results of the multivariable linear regression analysis of factors associated with COVID-19-related awareness, attitude, and behavior scores among travelers arriving in Saudi Arabia. We found that, compared with other nationalities, Indian, adjusted B 3.23 (95% CI: 2.51–3.94), and Pakistani, adjusted B 2.26 (95% CI: 1.51–3.02), were associated with higher awareness scores. Education was also positively associated with better awareness. Compared with secondary or lower education having a bachelor’s degree and a master’s or a higher level of education was associated with higher awareness scores, adjusted B 1.84 (95% CI: 1.19–2.48) and adjusted B 1.37 (95% CI: 0.40–2.35), respectively. Income more than SAR 10,000 was associated with higher awareness scores, adjusted B 1.17 (95% CI: 0.35–2.00).

The only significant predictor of attitude score in the multivariable analysis was nationality. Indian and Pakistani nationalities were associated with lower attitude scores, adjusted B −1.77 (95% CI: −2.59–−0.94) and adjusted B −1.49 (95% CI: −2.36–−0.62), respectively. The female gender was associated with better behavior, adjusted B 0.45 (95% CI: 0.25–0.65). Being Indian compared with other nationalities was associated with significantly higher behavior scores, adjusted B 0.63 (95% CI: 0.33–0.92).

Satisfaction of travelers with infection control measures at airports is summarized in Table 3. In the domain of transportation, the highest number of participants (97.6%) were satisfied with distancing in the intra-terminal buses. Overall satisfaction with transportation was about 94%. The highest satisfaction in the domain of lounges and corridors was about the presence of distance markings (99.7%) and cleanliness (97.7%). While the lowest satisfaction was with the availability of health messages in different languages (26%) and the behavior of travelers about following social distancing at luggage belts (31%). Overall satisfaction with lounges and corridors was 96%. At the counters, the highest satisfaction was with cleanliness (99%) while the lowest was with compliance of people with social distancing (66%). Overall satisfaction with counters was 98%. Concerning airport staff, all (100%) of the participants reported that staff wore masks. Overall satisfaction with airport staff was about 97%.

## 4. Discussion

Human mobility across countries has been found to be a substantial contributing factor to virus spread, in which air travel played a predominant role in rapid virus transmission [28]. Information about disease, preventive behavior, and risk perceptions among travelers are crucial for the control of COVID-19 and other emerging diseases such as monkeypox [29]. Therefore, exploring travelers’ awareness and behaviors toward COVID-19 prevention and their satisfaction levels regarding precautions adopted by air travel management authorities was essential and timely. Although the majority of the knowledge, attitude, and practices’ (KAP) surveys during the COVID-19 pandemic era were carried out in different population groups [13,14], our study is unique in terms of assessing the awareness, attitude, behaviors, and satisfaction levels among expatriates in KSA who were traveling back to Saudi Arabia from their home countries.

Awareness scores related to COVID-19 were adequate as reported by our study participants. The finding is supported by studies that explored knowledge about COVID-19 among travelers [30], as well as among the general population [31,32]. The evidence regarding a good knowledge of COVID-19 was logical, as the infection was not confined to the local geographical boundaries but was a pandemic with no treatment and high mortality rates globally and the digital media all around the world was the strongest source to spread the information [33]. Moreover, higher education added more value and made a difference in gaining knowledge quickly about COVID-19 and its preventive aspects [34].

Attitude and intention to act are vital measures for translating knowledge into practice; however, a contradictory finding has been shown in our study as lower attitude scores were revealed among Indian and Pakistani participants in our study. Prapaso et al. also found a neutral attitude among Thai travelers towards COVID-19, which has justified the finding by explaining the neutral and controlled situation of the pandemic in Thailand when data were obtained [30]. Our study participants might have observed the ease of restrictive measures before travel in their home countries due to control of the COVID-19 situation, therefore low attitude and risk perception was noted at the time of data collection. Furthermore, this low attitude score among Indian and Pakistani nationals could also be due to the overall low level of compliance with the restrictions in their countries, as they might have been stuck in their home countries since the beginning of the pandemic. Hence, in contrast to our study findings, other studies found positive attitude scores towards COVID-19 during the pandemic era among travelers [21] and other groups as well [35,36]. Nevertheless, evidence suggests a strong association of attitude with preventive behavioral intentions towards COVID-19 [37].

In terms of preventive behaviors, our study findings revealed good behavior scores among travelers. This finding is comparable to a survey conducted among travelers in Thailand, which showed adequate practice scores during the peak of the pandemic [30], and the results are justified by a positive association of risk perception with the preventive behavior that was studied in Indonesia [37]. Notably, the female gender in our study was found to be associated with better practices for preventing the infection than males. Our finding is comparable to results from various studies among travelers [30] and general population that showed that females were more likely to practice preventive measures. This is likely due to their higher risk perceptions and lower confidence in health systems [36,38,39,40]. Evidence also revealed that pre-travel advice and counseling was substantially associated with better practice scores for COVID-19 prevention [30]. The literature also suggested that the fear of contracting the COVID-19 infection was substantially associated with compliance with preventive practices [41]. This may explain our study finding of low attitude scores but higher practices’ scores that might be a result of fear of the disease.

The age and behavior scores for COVID-19 prevention in our study were inversely related in the bivariate correlation. However, this was not significant in the multivariable analysis. On the contrary, the recent evidence suggests that older people are at higher risk of infection and more severe disease and complications, perceive more threat from COVID-19 infection, and acquire more adequate behaviors than the younger age groups [38,42,43]. Education, occupation, and income are correlated and coherent to the higher awareness; our study also reported that the travelers who received a higher education and had a good income had better knowledge about the pandemic. This may reflect that they have been exposed to social media more and take interest in gaining enough information about COVID-19 as they intend to travel across countries [33]. Consistent comparable results from other studies revealed that higher education and income have a positive association with higher knowledge scores about the virus and better practicing skills among travelers as well as in other groups [30,35]. A study conducted in the United States revealed that the participants, who were poor, black, and had low literacy levels, were less worried and less prepared for the pandemic [44].

The satisfaction level among the travelers at the airport in our study was higher for distancing at terminals, distance markings in lounges and corridors, overall cleanliness, and full compliance of airport staff for wearing masks. The findings are comparable with the studies conducted to analyze the compliance of essential safety measures at airports which revealed that the steps taken such as social distancing, face mask use, disinfection and cleanliness, temperature and symptoms check, and COVID-19 infection testing contributed significantly in increasing the satisfaction level among passengers at the airports [20,23,45]. Passengers at Indian airports reported lower satisfaction with preventive measures’ implementation such as wearing masks, gloves, and social distancing [8]. While in our study, compliance with distancing at the luggage belt and counters and limited availability of health messages in different languages were found to be unsatisfactory. Crowd management and queuing to control a crowd is very challenging; however, studies have shown that queuing time is a potential influencing factor for the higher satisfaction level among flight passengers [45,46]. Another research carried out among frequent flyers revealed that passengers agreed to receive complimentary sanitizer and masks, along with additional information about prevention aspects, so that they can feel safer while flying [21].

Our study carries some strengths: (1) conducting research among expatriates for assessing their awareness, attitude, behaviors, and satisfaction level, who might be the potential carriers for virus transmission, has added value to the literature. (2) determining mean scores for awareness, attitude, and behaviors gives a more meaningful interpretation of results. There are a few limitations linked to our study that need to be considered while interpreting and implying the study findings. First of all, awareness, attitude, and behaviors were self-reported and were not observed objectively on site, particularly for the practicing-related part, which might produce a reporting bias. Second, although the study was designed early during the pandemic, due to continued restrictions on air operations data were collected when the pandemic was almost controlled and restrictive measures were somewhat eased, which could have resulted in better awareness scores. Third, we included only non-Saudi travelers, which limits study generalizability. However, during the study period, it was likely that Saudi travelers would constitute a smaller proportion of arriving passengers, as Saudi nationals who were stuck outside KSA during pandemic were brought back through special flights early during the pandemic. Fourth, in our sample, the female participants were 24%, which is slightly lower than the estimated 33% females among expatriates in KSA [47]. However, we assume this to have minimal effects on the generalizability of our results. Finally, in our sample, Indian and Pakistani nationals were overrepresented, which might affect the generalizability of the results on the total expatriate population of Saudi Arabia. Nonetheless, Indians and Pakistanis are the two largest non-Saudi nationals in Saudi Arabia.

## 5. Conclusions

This study found that the travelers had better awareness (16.3/22) and attitudes (18.3/24), while behavior scores (3.2/5) were relatively low. This calls for actions to improve and facilitate positive practices through behavior change communication. In the regression analysis, the nationality of the travelers was found to be a significant predictor of awareness, attitude, and behaviors. There is need for further research to explore the underlying factors that moderate the association of nationality with COVID-19 related awareness, attitude, and behaviors among travelers. Regarding satisfaction with preventive measures at airports, we found that there was high satisfaction in most of the domains studied. However, availability of health messages in different languages and proper distancing at luggage belts and counters were areas of low satisfaction. Poor satisfaction with distancing at counters and luggage belt also indicates poor safety behaviors of travelers in these areas. Airport authorities at international airports should provide multilingual health messages in order to improve the compliance of travelers from different nationalities. Furthermore, there is also a need to improve crowd management at the luggage belts and counters through proper management of the disembarkment of passengers and strict monitoring of travelers’ behaviors in high crowd areas.

## Figures and Tables

**Table 1 tropicalmed-07-00435-t001:** Socio-demographic characteristics and COVID-19-related awareness, attitude, and behaviors of travelers arriving in KSA (*n* = 633).

Value	N (%)	Awareness Mean (SD)	Attitude Mean (SD)	Behavior Mean (SD)
**Age (Mean SD)**	36.26 (8.7)	−0.67 *	0.10 *	−0.081 *
*p*-value		0.093	0.796	0.045
**Gender**				
Male	477 (76%)	16.0 (3.0)	18.2 (3.1)	3.1 (1.1)
Female	151 (24%)	17.2 (2.1)	18.6 (2.2)	3.6 (1.0)
*p*-value		<0.001	0.05	<0.001
**Marital status**				
Ever married	550 (86.9%)	16.3 (2.8)	18.3 (2.9)	3.2 (1.0)
Never married	83 (13.1%)	16.3 (2.9)	18.0 (3.1)	3.1 (0.99)
*p*-value		0.91	0.41	0.63
**Nationality**				
Indian	352 (55.6%)	16.9 (2.5)	18.0 (2.8)	3.4 (0.9)
Pakistani	215 (34%)	16.3 (2.4)	18.3 (3.0)	3.0 (1.2)
Others	66 (10.4%)	13.0 (3.4)	19.7 (2.7)	2.6 (0.8)
*p*-value		<0.001	<0.001	<0.001
**Educational level**				
Secondary school or lower	122 (19.3%)	14.6 (3.3)	18.8 (2.6)	2.9 (1.0)
High school	222 (35.1%)	15.8 (2.9)	18.1 (3.2)	3.0 (1.0)
Bachelor’s degree	209 (33%)	17.5 (1.9)	17.9 (2.9)	3.4 (0.9)
Master’s degree or higher	75 (11.8%)	17.4 (2.1)	19.1 (2.2)	3.5 (1.2)
*p*-value		<0.001	0.002	<0.001
**Employment Status**				
Unemployed	40 (6.3%)	16.4 (2.1)	19.1 (2.7)	3.3 (1.1)
Private sector	518 (81.8%)	16.2 (3.0)	18.1 (2.96)	3.1 (1.0)
Governmental sector	68 (10.7%)	17.2 (2.4)	19.0 (2.5)	3.6 (1.2)
*p*-value		0.024	0.017	0.004
**Monthly Income**				
Less than SAR 3000		15.6 (3.2)	18.3 (3.1)	3.1 (0.9)
SAR 3001–10,000		16.6 (2.7)	18.1 (2.8)	3.2 (1.1)
More than 10,000		17.4 (2.0)	18.4 (2.6)	3.4 (1.2)
*p*-value		<0.001	0.551	0.079

* Pearson correlation coefficient.

**Table 2 tropicalmed-07-00435-t002:** Predictors of COVID-19-related awareness, attitude, and behaviors of travelers arriving in KSA.

Value	Awareness		Attitude		Behaviors	
	Beta	95% CI	Beta	95% CI	Beta	95% CI
**Age (Mean SD)**	−0.01	−0.04–0.01	-	-	−0.01	−0.02–0.00
**Gender**						
Male	1	1	-	-	1	1
Female	0.95	0.42–1.48	-	-	0.45	0.25–0.65
**Marital status**						
Ever married	-	-	-	-	1	1
Never married	-	-	-	-	−0.19	−0.46–0.08
**Nationality**						
Indian	3.23	2.51–3.94	−1.77	−2.59–−0.94	0.63	0.33–0.92
Pakistani	2.26	1.51–3.02	−1.49	−2.36–−0.62	0.11	−0.20–0.42
Others	1	1	1	1	1	1
**Educational level**						
Secondary school or lower	1	1	1	1	1	1
High school	0.51	−0.07–1.08	−0.20	−0.87–0.47	0.02	−0.22–0.25
Bachelor’s degree	1.84	1.19–2.48	−0.22	−0.96–0.52	0.30	0.03–0.56
Master’s degree or higher	1.37	0.40–2.35	0.97	−0.15–2.09	0.21	−0.16–0.58
**Employment Status**						
Unemployed	0.52	−0.57–1.60	0.11	−1.14–1.35	-	-
Private sector	0.54	−0.17–1.25	−0.80	−1.62–0.03	-	-
Governmental sector	1	1	1	1	-	-
**Monthly Income**						
Less than SAR 3000	1	1	1	1	1	1
Sar 3001–10,000	0.35	−0.16–0.86	−0.42	−1.00–0.15	−0.08	−0.28–0.12
More than 10,000	1.17	0.35–2.00	−0.80	−1.72–0.12	0.24	−0.09–0.57

**Table 3 tropicalmed-07-00435-t003:** Satisfaction of travelers with infection control measures at airports in Saudi Arabia during the COVID-19 pandemic.

Domain	*n* (%)
**Transportation**	
Maintaining distance in intra-terminal buses	618 (97.6)
Hand sanitizers are available on intra-terminal buses	483 (76.3)
Baggage carts are clean	587 (92.7)
Baggage carts are being sanitized	240 (37.9)
Overall satisfaction with transportation	594 (93.8)
**Lounges and corridors**	
Cleanliness	618 (97.6)
Availability of hand sanitizers	483 (76.3)
Health messages are displayed	591 (93.4)
Health messages are available in different languages	164 (25.9)
Distance markings are present	631 (99.7)
Seating arrangements are in such a way that a reasonable distance is maintained	611 (96.5)
Luggage belt is properly organized to maintain distance	408 (64.5)
People are following distancing while waiting for luggage	196 (31)
Overall satisfaction with lounges and corridors	609 (96.2)
**Counters**	
Cleanliness	627 (99.1)
Availability of hand sanitizers	537 (84.8)
Distance markings are present	625 (98.7)
People are following distancing while in the queue	418 (66)
Overall satisfaction with counters	620 (97.9)
**Airport Staff**	
Wearing mask	633 (100)
Wearing gloves	413 (65.2)
Maintaining distance	578 (91.3)
Asking people to wear a mask	578 (91.3)
Helping people to maintain distance	381 (60.2)
Overall satisfaction with airport staff	613 (96.8)

## Data Availability

The data presented in this study are available on request from the corresponding author.

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
