# Peer review of "COVID-19-Related Awareness and Behaviors of Non-Saudi Travelers and Their Satisfaction with Preventive Measures at Saudi Airports"

_tropicalmed, 2022, doi:10.3390/tropicalmed7120435_

Round 1

Reviewer 1 Report

Alsaeed et al. conducted a cross-sectional survey to explore COVID-related awareness, attitude, behaviour, and satisfaction of travellers arriving in Saudi Arabia airports. Although the results are interesting and could guide policy, there are certain issues that need to be addressed:

1.      The study does not seem to have included Saudi national or at least were not the majority (Table 1). The questionnaire was administered in three languages and not in English. These rises concern about the representativeness of the sample.

2.      The inclusion/exclusion of participants is unclear and the sentence in line 110 is incomplete.

3.      The authors very briefly mention that number of items asked for awareness, attitude, and behaviour, but do not provide more details of the questions and whether they used validated tool. The questions should be attached in the appendix.

4.      The process of approaching the participants and conducting the survey need to be described in detail, as wrong procedures could lead to social desirability bias.

5.      The selection of the variables into the multivariable regression model needs to be described and whether collinearity and other assumptions were tested. Collinearity (education level, employment status, monthly income) could lead to spurious/reversed associations such as the one reported by the authors of age and behaviour.

6.      The conclusions in the discussion and abstract (e.g., ‘At the airports, areas such as luggage belts and counters need special attention prevent potential transmission of infections’) do not match the aims of the study nor are backed up by the results as transmission of disease was not assessed.

Author Response

We are thankful to the esteemed reviewers for their precious time and important comments/suggestions to improve the quality of manuscript. We have thoroughly revised the manuscript as per comments. Changes in the manuscript are highlighted in yellow. Following is the point-by-point response to each of the comments of reviewers.

Alsaeed et al. conducted a cross-sectional survey to explore COVID-related awareness, attitude, behaviour, and satisfaction of travellers arriving in Saudi Arabia airports. Although the results are interesting and could guide policy, there are certain issues that need to be addressed:

  1. The study does not seem to have included Saudi national or at least were not the majority (Table 1). The questionnaire was administered in three languages and not in English. These rises concern about the representativeness of the sample.

Response: Thanks for the comment. As mentioned in the eligibility criteria “Non-Saudi….”. This study included non-Saudi travelers arriving to Saudi Arabia after resumption of flight operations.  We have mentioned the same in the limitations of our study.  

The inclusion/exclusion of participants is unclear and the sentence in line 110 is incomplete.

Response: Thanks for the comment. We have revised the section on selection criteria and hope that it is better than earlier. The incomplete sentence was mistakenly deleted text which was overlooked. The sentence has been completed now.

  1. The authors very briefly mention that number of items asked for awareness, attitude, and behaviour, but do not provide more details of the questions and whether they used validated tool. The questions should be attached in the appendix.

Response: Thanks for valuable comment. The questionnaire was adopted from previous studies in Saudi Arabia which have validated the questionnaire. References of those studies have already been provided. Concerning the more details on items, we restricted details to main ideas within each of the domains for the sake of brevity. As per suggestion, we have uploaded questionnaire as supplementary material for readers full understanding.

  1. The process of approaching the participants and conducting the survey need to be described in detail, as wrong procedures could lead to social desirability bias.

Response: We have provided further details about the selection procedure.

  1. The selection of the variables into the multivariable regression model needs to be described and whether collinearity and other assumptions were tested. Collinearity (education level, employment status, monthly income) could lead to spurious/reversed associations such as the one reported by the authors of age and behaviour.

Response: Thanks for the comments. Details on developing final multivariate models have been added about selection of variables for multivariate models and collinearity assessments. Relevant references have also been added. Regarding association of age with behaviors, there was very weak correlation in the bivariate analyses which was not significant anymore in the multivariate model. The same has been elaborated within discussion.

  1. The conclusions in the discussion and abstract (e.g., ‘At the airports, areas such as luggage belts and counters need special attention prevent potential transmission of infections’) do not match the aims of the study nor are backed up by the results as transmission of disease was not assessed.

Response: We agree that the transmission of infection was not assessed in the current study. However, close space among the travelers may serve as ‘potential’ source of transmission of infection. Therefore, in the conclusions, we mentioned ‘potential’ as we did not assess actual transmission.  

Reviewer 2 Report

Interesting study thanks - just a few things 

You need to get a native english speaker to correct it 

eg Line 12  - missing preposition 

Airports could serve as a  hotspot for the spread of COVID-19 infection.

and it goes on - very distracting to a native English speaker

also 

Line 110 stops without finishing the sentence 

More should be made of the 24% female  reply rate in the disadvantages 

I would be interested to know if passengers understood that sanitising etc is not that effective for COVID and that it is poor ventilation that is the primary driver of transmission 

Author Response

We are thankful to the esteemed reviewers for their precious time and important comments/suggestions to improve the quality of manuscript. We have thoroughly revised the manuscript as per comments. Changes in the manuscript are highlighted in yellow. Following is the point-by-point response to each of the comments of reviewers.

Interesting study thanks - just a few things 

You need to get a native english speaker to correct it 

eg Line 12  - missing preposition 

Airports could serve as  hotspot for the spread of COVID-19 infection.

and it goes on - very distracting to a native English speaker

 Response: Thanks for identifying issues with the language. We have revised manuscript thoroughly for the English language and hope that it meets the standards now.

Line 110 stops without finishing the sentence 

 Response: The incomplete sentence was mistakenly deleted text which was overlooked. The sentence has been completed now.

More should be made of the 24% female  reply rate in the disadvantages 

 Response: According to the general authority of statistics of Saudi Arabia, the female among expatriate population constitutes about 33%. In our sample, this proportion was 24% which is lower than the actual proportion. However, we assume that this would not affect the generalizability of the results to a great extent. We have stated the same in limitations.

I would be interested to know if passengers understood that sanitising etc is not that effective for COVID and that it is poor ventilation that is the primary driver of transmission 

Response: Thanks for the comment. Unfortunately, we did not have variables related to this idea.

Reviewer 3 Report

It was a pleasure to review the original article titled “COVID-19-related awareness and behaviors of travelers and their satisfaction with preventive measures at Saudi airports” Ms ID “tropicalmed-2053370” submitted for publication in “TropicalMed (ISSN 2414-6366)”.

The paper is interesting and pleasant to read. However, if you don’t mind.  I have some comments for the authors, there are some things that should be corrected to ensure that the manuscript is suitable for publication:

  1. Even if the paper is very pleasant to read, there are some grammatical mistakes and run-on sentences that must be corrected. I suggest to the authors to carefully check the grammar of their manuscript. Please try to make a revision by a native English, a specialist or an English editing service
  2. Line 20, maybe you should add a value to support the significance of nationality (at least a p value)
  3. Line 30, please start by writing the full term eg. “Coronavirus disease (COVID-19)” instead of “COVID-19” as it is the first time it appears in your text.
  4. Line 31, please add a reference in the end of the sentence.
  5. I think you have to add at least between 1-3 short sentences about the COVID-19 pandemic in general (infections/deaths, preventive measures, socio-economic and psychological effects…etc) before writing about your specific thematic. In addition, there is a lack of references in the paper, here are some papers that you must add to support your study:

* https://doi.org/10.1016/j.dhjo.2022.101278 

* https://doi.org/10.3390/healthcare1007134  

* https://doi.org/10.1016/j.landusepol.2021.105772

* https://doi.org/10.1080/21582041.2021.1975809

L      6. Line 39, Please add some details about vaccination campaign and side effects in the word and in Saudi Arabia, here are some papers to add:

* https://doi.org/10.1007/978-3-030-92901-5_9

* https://doi.org/10.3390/tropicalmed7040060

* https://doi.org/10.3390/vaccines10111781

* https://doi.org/10.1111/aspp.12643

7     7.    Line 60, please change the term “done”

8     8.      Line 81, please start a paragraph specifically dedicated to the aims of your study.

9    9.      Line 89, please specify the day.

1    10.   Line 110, participants were ???

1    11.   Why didn't you use a questionnaire in English directly? normally it's the second official language of the country, no?

1    12.   Line 133, please add the full name of SPSS, details (IMB, Addinsoft)…etc. and add a citation.

1    13.   Line 215, please delate extra parentheses

1    14.   Line 250, please change the expression “of note”.

1    15.   Line 260, you can specify that age/older age has always been associated with disease infection and/or severity.

1    16.   Maybe it’s better to start a new paragraph for both strengths and limitations of your study, you can even add specific titles for each section

1    17.   In general, your results and discussion sections were well handled, it’s great.

1    18.   Line 307, please rephrase or change by another sentence as it is not very representative.

1    19.   Your conclusion section is weak, I think it should be rewritten, the readers must understand your whole study just by reading your conclusion.

      Best Wishes.

Author Response

We are thankful to the esteemed reviewers for their precious time and important comments/suggestions to improve the quality of manuscript. We have thoroughly revised the manuscript as per comments. Changes in the manuscript are highlighted in yellow. Following is the point-by-point response to each of the comments of reviewers.

It was a pleasure to review the original article titled “COVID-19-related awareness and behaviors of travelers and their satisfaction with preventive measures at Saudi airports” Ms ID “tropicalmed-2053370” submitted for publication in “TropicalMed (ISSN 2414-6366)”.

The paper is interesting and pleasant to read. However, if you don’t mind.  I have some comments for the authors, there are some things that should be corrected to ensure that the manuscript is suitable for publication:

  1. Even if the paper is very pleasant to read, there are some grammatical mistakes and run-on sentences that must be corrected. I suggest to the authors to carefully check the grammar of their manuscript. Please try to make a revision by a native English, a specialist or an English editing service

Response: Thanks for the comment. We have revised manuscript for English language.

  1. Line 20, maybe you should add a value to support the significance of nationality (at least a p value)

Response: Thanks for the suggestion. However, due to word count restriction on abstract length, we restricted the information to the names of variables only without mentioning the value of coefficient or p-values.

  1. Line 30, please start by writing the full term eg. “Coronavirus disease (COVID-19)” instead of “COVID-19” as it is the first time it appears in your text.

Response: We have revised as per comment.

  1. Line 31, please add a reference in the end of the sentence.

Response: Reference has been added as per comment.

  1. I think you have to add at least between 1-3 short sentences about the COVID-19 pandemic in general (infections/deaths, preventive measures, socio-economic and psychological effects…etc) before writing about your specific thematic. In addition, there is a lack of references in the paper, here are some papers that you must add to support your study:

* https://doi.org/10.1016/j.dhjo.2022.101278 

* https://doi.org/10.3390/healthcare1007134  

* https://doi.org/10.1016/j.landusepol.2021.105772

* https://doi.org/10.1080/21582041.2021.1975809

Response: Thanks for the comments and recommending article. For the sake of brevity, we focused on main theme of our manuscript. However, as per suggestion we have added little more details on COVID-19 pandemic along with references.

L      6. Line 39, Please add some details about vaccination campaign and side effects in the word and in Saudi Arabia, here are some papers to add:

* https://doi.org/10.1007/978-3-030-92901-5_9

* https://doi.org/10.3390/tropicalmed7040060

* https://doi.org/10.3390/vaccines10111781

* https://doi.org/10.1111/aspp.12643

Response: Thanks for the suggestion. However, we would respectfully disagree with adding details about COVID-19 vaccine campaigns, acceptance, and adverse effects. Vaccination does not relate directly to the main idea of paper and may lead to unnecessarily lengthy introduction and redundancy.

7     7.    Line 60, please change the term “done”

Response: The term has been changed.

8     8.      Line 81, please start a paragraph specifically dedicated to the aims of your study.

Response: Aims have been presented as separate paragraph.

9    9.      Line 89, please specify the day.

Response: The dates have been specified.

  • Line 110, participants were ???

Response: The incomplete sentence was mistakenly deleted text which was overlooked. The sentence has been completed now.

  • Why didn't you use a questionnaire in English directly? normally it's the second official language of the country, no?

Response: We agree that the English is second official language in Saudi Arabia. However, our target population was expatriates, which included labors and other workers who may not be able fully understand the questionnaire in English. For this reason, we translated questionnaires so as to obtain valid responses from the participants.

  • Line 133, please add the full name of SPSS, details (IMB, Addinsoft)…etc. and add a citation.

Response: Full name and details of software used for analysis have been added.

  • Line 215, please delate extra parentheses

Response: Extra parenthesis has been deleted.

  • Line 250, please change the expression “of note”.

Response: The expression has been changed.

  • Line 260, you can specify that age/older age has always been associated with disease infection and/or severity.

Response: The suggestion has been incorporated along with reference.

  • Maybe it’s better to start a new paragraph for both strengths and limitations of your study, you can even add specific titles for each section

Response: Thanks for the suggestion. However, we formatted these sections as per journal formatting guidelines.

  • In general, your results and discussion sections were well handled, it’s great.

Response: Thanks for the complements.

  • Line 307, please rephrase or change by another sentence as it is not very representative.

Response: Thanks for the comment. We have revised the conclusion.

  • Your conclusion section is weak, I think it should be rewritten, the readers must understand your whole study just by reading your conclusion.

Response: Thanks for the comment. We have revised the conclusion.

Round 2

Reviewer 3 Report

The authors adressed almost all my comments even if they ignored some of them. However, the paper remains interesting and is suitable for publication.

Good luck.

Author Response

We are thankful to the esteemed reviewer for reviewing our manuscript for the second time. 

The authors adressed almost all my comments even if they ignored some of them. However, the paper remains interesting and is suitable for publication.

Good luck.

Response: We are also glade that the reviewer is largely satisfied with our revisions and responses and grading manuscript 'suitable for publication'.